SCIENCE FORUM

# Building a community to engineer synthetic cells and organelles from the bottom-up

**Abstract** Employing concepts from physics, chemistry and bioengineering, 'learning-by-building' approaches are becoming increasingly popular in the life sciences, especially with researchers who are attempting to engineer cellular life from scratch. The SynCell2020/21 conference brought together researchers from different disciplines to highlight progress in this field, including areas where synthetic cells are having socioeconomic and technological impact. Conference participants also identified the challenges involved in designing, manipulating and creating synthetic cells with hierarchical organization and function. A key conclusion is the need to build an international and interdisciplinary research community through enhanced communication, resource-sharing, and educational initiatives.

OSKAR STAUFER*, JACQUELINE A DE LORA, ELEONORA BAILONI, ALISINA BAZRAFSHAN, AMELIE S BENK, KEVIN JAHNKE, ZACHARY A MANZER, LADO OTRIN, TELMO DÍEZ PÉREZ, JUDEE SHARON, JAN STEINKÜHLER, KATARZYNA P ADAMALA, BRUNA JACOBSON, MARILEEN DOGTEROM, KERSTIN GÖPFRICH, DARKO STEFANOVIC, SUSAN R ATLAS, MICHAEL GRUNZE, MATTHEW R LAKIN, ANDREW P SHREVE, JOACHIM P SPATZ* AND GABRIEL P LÓPEZ*

**\*For correspondence:**
oskar.staufer@mr.mpg.de (OS);
Joachim.Spatz@mpimf-heidelberg.mpg.de (JPS);
gplopez@unm.edu (GPL)

**Competing interest:** The authors declare that no competing interests exist.

## Philosophy of the field and past achievements

Physicists and engineers traditionally focus on the non-living world and apply model systems with reduced complexity to capture the essentials of a living entity, and to gain mechanistic insights into higher-order processes on a micro-, meso- and macroscale. This approach is also becoming more popular in the life sciences, where a 'learning-by-building' strategy is used to design and construct synthetic cells and organelles of reduced but defined complexity (*Jia and Schwille, 2019*; *Ausländer et al., 2017*). Early national and transnational initiatives, such as the Max Planck School 'Matter to Life' (https://mattertolife.maxplanckschools.org), the Build-A-Cell research coordination network (https://www.buildacell.org/), the Building a Synthetic Cell (BaSyC) research program (https://www.basyc.nl/about-basyc/), and the International Genetically Engineered Machine competition (https://igem.org/Competition), have been established to advance training, research and collaboration in this exciting new field.

Cells are the basic units of life. But their intricate structure and the tightly orchestrated interplay of individual molecular components within cells are far from basic. Most cellular phenomena are not understandable through intuition but require complex analytical systems to provide a mechanistic description of the processes forming living matter. Yet, within the complexity of living cells hide the answers to some of the most fundamental questions in the life sciences, from the emergence of life and the transition from inanimate matter to life, to the development and cure of diseases.

**Figure 1.** Research on engineering synthetic cells and organelles, as represented at SynCell2020/21, covers a wide range of experimental systems including engineered cells created using standard transformation techniques, minimal cells, membrane-encapsulated synthetic cells, and all of the above with the possible inclusion of engineered membraneless organelles that produce hierarchical structures. The common objective of the field is to engineer synthetic structures with defined complexity to mimic biological systems on multiple length scales. The creation and characterization of these experimental systems draws on a wide variety of interdisciplinary inputs, including biological cells, biochemical components (such as cell-free TXTL extracts), and control programs that encode desired behavior in a variety of formats. In addition, a broad range of computational and experimental tools are required.

The online version of this article includes the following figure supplement(s) for figure 1:

**Figure supplement 1.** The International Conference on Engineering Synthetic Cells and Organelles was originally scheduled to take place in 2020 in Santa Fé in the United States with 150 participants.

These compelling and profoundly difficult questions reflect a vision for the future of the field as expressed by the SynCell2020/21 early-career panelists. The philosophical and ethical considerations underlying these questions, e.g., the misuse of synthetic cells for biological warfare, the impact of synthetic cells on natural environments, or the unpredictable nature of completely new life forms, are notable for their contrast with technological and engineering-focused objectives (*Gallup et al., 2021*).

Using principles from biology and engineering, interdisciplinary research teams have applied synthetic cells to construct materials and hierarchical structures with life-like properties that recreate essential features of living cells and reach beyond the capabilities of natural cells (*Figure 1*; *Elani et al., 2018*; *Krinsky et al., 2018*; *Rampioni et al., 2018*; *Staufer et al.,*

*2021a*; *Tian et al., 2019*). Cellular functions entail a diverse spectrum of modular components that regulate cell behavior. These include the ability to process chemical and physical signals from extracellular stimuli (information processing and decision making), cellular motions and adaptation to the environment (molecular adaptivity), cellular replication through division (proliferation) and nutrient uptake and garbage disposal (energy homeostasis). These abilities have helped to guide the functional design of synthetic cells and their equipment with modules that allow them to interact with the extracellular environment and to discriminate extracellular signals. For instance, extracellular morphogen sensing systems have been implemented in protocells that are able to change their structure and to discriminate between uni- and counter-directional morphogen gradients (*Tian et al., 2019*). Tactile behavior in

the form of chemo- and phototaxis has also been engineered into synthetic cell surrogates (*Bartelt et al., 2018*; *Pan et al., 2019*).

Broadly speaking, there are two primary approaches to construct low-complexity synthetic cells: top-down and bottom-up (*Ausländer et al., 2017*). Top-down approaches make use of existing living cells and sequentially remove individual components such as single genes (*Lachance et al., 2019*). This process can be iterated until reaching the absolute lowest point of complexity required for a cell to live. Analogous to synthetic lethality experiments in model organisms, top-down approaches provide descriptive insights into which parts of a cell are most crucial. However, it can be harder to obtain a systems-level understanding of how the parts work together.

In contrast, the bottom-up approach rationally combines non-living molecules in an understand-by-design approach to activate and exhibit the behaviors of living cells within artificial structures (*Buddingh' and van Hest, 2017*). A common defining element of cellular life forms is the ability to replicate a compartmentalized information-storing and self-sustaining, out-of-equilibrium system that manifests itself in specific characteristics, which can be selected in an evolutionary process (*Benner, 2010*; *Damiano and Stano, 2020*; *Porcar et al., 2011*). This could be achieved by engineering a compartmentalized entity that exhibits a metabolism for reproduction purposes and environmental adaptation. For example, a lipid membrane vesicle that can grow and divide by catabolizing external substrates and harboring DNA-encoded genes that specify the enzymes required for catabolism and reproduction. The advantage of the bottom-up approach is that every component of the created system can be located and defined in a quantitative manner, together with specified interactions between molecules.

But regardless of the approach – top-down or bottom-up – the knowledge gained from building synthetic cells has the potential to provide fundamental insights into life and to shape technologies of global impact, such as new vaccination strategies (*Dormitzer et al., 2013*), routes to overcome antibiotic resistance (*Wu et al., 2017*), new manufacturing pipelines for synthetic materials (*Le Feuvre and Scrutton, 2018*), and alternatives to petrochemicals (*Shih, 2018*).

Within the last decade, the field of building synthetic cells and organelles has achieved several major technological breakthroughs. A minimal synthetic cell consisting of only 473 genes, capable of metabolizing and reproducing, has been constructed using a top-down approach (*Hutchison et al., 2016*). With the objective of engineering a minimal synthetic cell synthetic chromosomes have been designed to generate artificial genetic blueprints for operating synthetic cell systems (*Greene et al., 2019*). In addition, droplet-based synthetic cells with an artificial photosynthetic metabolism that can bind $CO_2$ have been created, and synthetic cell systems for the scalable bio-production of natural plant products have also been built (*Miller et al., 2020*). Bioengineering concepts emerging from such studies have guided the implementation of new, application-focused technologies, e.g., cell-free expression systems. And most recently, the advanced synthetic genetic codes and technologies to rewire translational processes of protein production have provided the foundation for RNA-based SARS-CoV-2 vaccines (*Karikó et al., 2008*; *Mulligan et al., 2020*). Together with liposomal and lipid nanoparticle technologies, which program cellular uptake and processing of the RNA content, these vaccines have been a vital tool in the fight against COVID-19 (*Park et al., 2021*).

Exploring the fundamentals of life as illustrated in these brief examples requires diverse skill sets for designing and engineering experimental systems. It further needs an unbiased and creative mind with a strong interdisciplinary background to successfully integrate aspects of physics, chemistry, biology, and the information sciences. Similar to fields that explore artificial intelligence and neuromorphic computing (*Valeri et al., 2020*), students attracted to synthetic cell research often share enthusiasm and interests that go beyond their primary disciplines, incorporating aspects of philosophy and cognition within their research (e.g., the 'Synthetic Biology, Politics and Philosophy' workshop held at Bris-SynBio *Meacham and Prado Casanova, 2020*). This cross-fertilization provides unique opportunities for attracting young researchers into the field of engineering synthetic cells and organelles (see Box 1 and Box 2 in *Supplementary file 1* for selected quotations from young researchers who participated in the SynCell2021 Workshop).

## Recent research directions and bottlenecks

The creation of compartmentalized, cell-mimicking structures and the integration of coupled transcription-translation (TXTL) systems has seen substantial progress (*Garamella et al.,*

*2019*). However, considerable challenges, some seemingly paradoxical, remain. Many of these were highlighted at the recent SynCell2020/21 conference (see *Figure 1—figure supplement 1* and Tables S1–S5 in *Supplementary file 1* for listings of the conference program and links to recordings) and have been extensively reviewed recently (*Gallup et al., 2021*; *Meng and Ellis, 2020*).

One of the most demanding challenges remains the coupling of information-encoding systems with self-replicating cell-like entities (*Walker et al., 2017*), which can be framed in terms of von Neumann´s abstract generality about the logic of cell-like, self-replicating automata (*von Neumann, 1966*) Such entities require both a mechanism to copy the cellular architecture and functionalities that allow copying of genetic information specifying cellular structure and function. Such units need molecular systems that link the functional parts of a synthetic cell to a decoding mechanism that reads the genetic instructions required to autocatalytically build a new cell. They also require a molecular module that copies and reinserts a transcript of the (genetic) instruction into the synthetic daughter cell (*Olivi et al., 2021*). This is the logical basis of self-reproduction.

The first steps towards the synthetic construction of such systems were presented at the conference. DNA-encoded genetic systems represent just one implementation of self-reproduction, leaving ample room for designing alternatives to fulfil the basic conditions for a 'living', synthetic cell (*Dreher et al., 2021*; *Otrin et al., 2021*). In addition, synthetic cells will also require control programs to orchestrate the interconnected processes of sensing, response and metabolism necessary for replication and other processes relevant for life-like behavior (*Lakin and Stefanovic, 2016*; *Li et al., 2021*; *Steinkühler et al., 2020*).

Important progress reported at the meeting also included the engineering of synthetic structures with hierarchical organization inspired by eukaryotic life forms. Several implementations of such systems, such as hierarchical intrinsically disordered protein and nucleic acid droplets generated within synthetic cell-like compartments, were presented (*Simon et al., 2017*; *Pérez et al., 2021*). These efforts are aimed at deconvolving the organizational principles of life, including the highly dynamic cross-scale architecture of eukaryotic and multicellular organisms, most apparent during embryogenic development and tissue regeneration.

How the structural organization of subcellular, cellular and tissue components is hard-wired and how degrees of plasticity in respective structures are regulated, are problems of such immense complexity for which approaches including multi-centered global screening efforts have not been able to resolve the underlying principles. New methods based on in vitro synthetic model systems of lower complexity may provide new insights into these processes.

For example, a pivotal driving force behind tissue organization consists of genetic feedback loops based on reaction-diffusion processes and hysteresis, as first proposed by Alan Turing in his work on the chemical basis of morphogenesis in the mid-twentieth century (*Turing, 1997*). This is a prime example of how reductionist approaches in the form of precisely defined models can be applied to the study of complex behaviors in biological systems. Researchers in synthetic biology have recently recreated Turing patterns from protein-based systems and used these to study decision-making during cellular organization and symmetry breaking (*Glock et al., 2019*). This underscores both the fundamental impact of the questions asked in the field and their longstanding relevance that argues for the need to pursue novel theoretical, computational and experimental approaches by unbiased young scientists working in integrative research communities.

Other approaches have contributed insights into the spatio-temporal dynamics and organization principles of membrane-less organelles (*Pérez et al., 2021*). Until recently, studying such dynamic structures in living cells has mostly been limited by a lack of perturbation capabilities and the undefined chemical environment within the cytosol. However, through in vitro reconstitution of intrinsically disordered protein/nucleic acid systems in isolated low-complexity environments, quantitative insights into the molecular and thermodynamic principles needed for assembly and homeostasis of phase-separated organelles has been achieved (*Linsenmeier et al., 2019*). Understanding the hierarchical organization principles of life will ultimately enable the formulation of the principal laws of decision-making within living matter, and the basis of information processing and signal integration within cell collectives (*Staufer et al., 2018*; *Staufer et al., 2019*).

Engineering synthetic cells and organelles is not solely directed towards investigating biological principles, but also holds promise for practical applications. This offers the opportunity to explore an extensive technical repertoire. For

instance, microfluidic approaches have been developed to assemble synthetic cells with adjustable and tunable composition. Specifically, cell-sized compartments in the form of water-in-oil droplets that contain proteins, lipids or nucleic acids, provide means of engineering systems capable of genetic information processing and artificial genotype-to-phenotype coupling, where selection is exerted at the level of the synthetic cells´ phenotype but propagation of a selected trait depends on the relevant genetic information being carried forward (*Miller et al., 2020*; *Staufer et al., 2021b*; *Staufer et al., 2020*; *van Vliet et al., 2015*; *Weiss et al., 2018*). Such droplet-based approaches have also been adopted for lipid membrane engineering (*Lussier et al., 2021*; *Steinkühler et al., 2020*). Similarly, DNA nanotechnology has allowed to combine programmable molecular architectures with extrinsically controlled functions (*Bazrafshan et al., 2020*; *Jahnke et al., 2020*). In a combinatorial approach, integration of DNA nanoarchitectures with synthetic cells has synergized top-down and bottom-up strategies (*Jahnke et al., 2021*). These examples demonstrate the potential for technology innovation originating from the field.

Although the reductionist approach pursued in the field of engineering synthetic cells and organelles has proven to be powerful for several lines of research, it is also confronted with systematic limitations. Foremost, as an engineering approach that iteratively reduces the complexity of living systems, the construction of synthetic cells will always be subjected to the problem of 'hidden variables' (*Garcia et al., 2016*). Unobservable or unidentified molecular components of importance to natural systems might be underrepresented or absent in the recreated in vitro system. Such unknown variables could contribute to the stochastic behavior of a system, and biological phenomena could potentially be masked behind noise effects in synthetic cells and organelles.

Moreover, living cells cannot only be considered as the simple sum of their parts. Emergence in living systems is poorly defined and understood, and the possibility to capture this aspect by a modular approach remains to be evaluated. A second systematic limitation of the reductionist approach lies in the fact that reduced complexity does not directly imply increased understanding of a biological phenomenon. This is well illustrated by the creation of minimal bacterial genomes with only 473 genes (*Hutchison et al., 2016*). Despite the remarkably reduced complexity and size of this genome, 149 of the genes are still of unknown function. Addressing these limitations provides opportunities for ongoing research and collaborative efforts, even as progress continues in advancing understanding and functional applications of synthetic cells and organelles.

## Future perspectives and community

Engineering approaches set the stage for implementing synthetic functional modules capable of performing specific functions in synthetic cells. The successful combination of all individual elements within a single entity will be key to assemble synthetic living cells. This, in turn, requires integrated inter-laboratory solutions that allow for off-the-shelf unification of individual modules. Exchanging expertise between laboratories and universal module interfaces will be essential and will enable broad participation in the field.

Discussions during the SynCell2020/21 revealed several fundamental strategic frameworks and infrastructure that are needed to achieve such a successful integration of the global community. Firstly, in the interest of effective paywall-free knowledge transfer among researchers, open-access data repositories are needed. This will facilitate transfer of experimental protocols and sharing of data and blueprints for synthetic cell modules, effectively boosting access of interested students to the field. Moreover, standardization efforts that strive to provide universal norms for the design and assembly of synthetic cell modules and interfaces need to be developed. Specific implementations of such platforms could be arranged, inspired by the collaborative software development and version control platform GitHub, which has experienced community-wide acceptance within computer science and engineering fields. The Build-A-Cell network has embraced this approach and has begun to assemble such open-access repositories.

Secondly, engineering synthetic cells and organelles will be a model for new transcontinental educational modalities. SynCell2020/21 was organized by the National Science Foundation (USA) and the Max Planck Society (Germany). It also received support from national research programs, e.g., the Build-A-Cell network (USA-based) and the BaSyC program (Netherlands; *Figure 1—figure supplement 1*). Presentations by leading researchers in student-centered tutorials were a focus of the conference framework. Community-driven education programs for

specialized training in relevant domains (biology, physics, chemistry, microbiology, molecular biology, biophysics, computer science or ethics) will be key for equipping new generations with the necessary skills to successfully engineer living synthetic cells and organelles. International workshops and research summer schools will be important to develop a coherent, long-lasting community that fosters cross-generational collaborations among scholars.

At present, only a limited number of training and graduate programs focused on the engineering of synthetic cells and organelles have been established, such as the Max Planck School 'Matter to Life', the Cold Spring Harbor Laboratory Summer School on Synthetic Biology, and research programs supported by the US National Science Foundation 'Rules of Life' initiative. Their successful implementation will not only nurture the next generation of scientists but will also train a cohort of researchers to enable industrial applications. If possible, future events should be organized between all major research and teaching initiatives (*Figure 1—figure supplement 1*) to bring together the global expertise and emerging talent, and to promote a broad distribution of thought leadership across institutions as the field continues to grow and develop.

Lastly, following the learning-by-building approach, the field awaits a steadily growing demand for an integrated research infrastructure that provides computational power and specialized courses in molecular and genetic design. This includes molecular modelling of large-scale, whole-cell models to predict the interactions of engineered components with host cells. Access to advanced computational facilities and enhanced algorithms for simulations based on machine learning and optimization techniques, will greatly expand the scope for designing and constructing synthetic cells and organelles. Dedicated research centers, such as the Max Planck–Bristol Center for Minimal Biology, could provide such facilities, as well as technical support for the increasingly important administrative aspects to the field, including technology transfer procedures, handling of intellectual property issues, and curation of specialized genetic parts and molecular module libraries specified for the field (inspired by biobanks such as Addgene and large-scale gene and genome synthesis 'bio foundries', such as those funded by the Department of Energy in the US).

For all the proposed measures, commitment and support from funding bodies, political and regulatory authorities, and universities

with established teaching infrastructure, will be essential. Especially to successfully install a strategic, open-source platform for synthetic biology and student exchange programs, like the ones between the University of New Mexico and the Max Planck Society.

We also observe that connections to and inspiration drawn from other research communities will be important. For example, research advances addressing origin-of-life questions, the basic principles of life, and the exploration of eukaryogenesis, connect many scientific themes that arise in the study of synthetic cells and organelles. This was highlighted by the observation that many SynCell2020/21 participants are also active in these related communities. Furthermore, expanding the research community in synthetic cells to connect to these and other related scientific communities opens additional opportunities for research support, including that available from private and philanthropic foundations. Recent examples of such initiatives include the 'Life? – A Fresh Scientific Approach to the Basic Principles of Life' program, supported by the Volkswagen Foundation, and the 'Project on the Origin of the Eukaryotic Cell', sponsored jointly by the Gordon and Betty Moore Foundation and the Simons Foundation.

Specific measures will include a joint program between the research initiatives mentioned above, aimed towards continued organization of the SynCell conference as a think-tank for community building and research exchange. Moreover, the Build-a-Cell initiative has initiated several focused working groups, e.g., working towards collection and annotation of synthetic cell subtype components or towards establishing in silico modeling frameworks of synthetic cells with predictable behavior. These groups provide an optimal platform to develop future cross-scale-organized infrastructure that will be able to manage between different stakeholders from academia, industry and political authorities, while also serving as an advisory council representing the field's interest. Furthermore, concentrated efforts will be made to raise awareness in academic faculties and scientific societies towards the importance of establishing relevant teaching schemes in graduate and undergraduate programs.

A compelling model for developing and sharing modular tools across the diverse synthetic biology community can be found in the design of the original Unix multi-user operating system and subsequent community-driven, evolutionary development of Linux (*Raymond and Young,*

*2001*). Unix's 'graceful facilities' enabled users to create complex programs by using software 'pipes' to compose simple modules together: at the same time, the operating system was designed to facilitate communication among programmers as 'the essence of communal computing' (as seen in the video 'The Unix System: Making Computers Easier to Use'). Linux emerged from an unprecedented, worldwide open-source effort by volunteer programmers. These core values of streamlined, modular design and enthusiastic, open, collaborative development can similarly inform and shape progress in the synthetic cell community.

## Conclusions

SynCell2020/21 demonstrated remarkable engagement of a large and geographically diverse community, and potential for global collaboration and transcontinental knowledge-sharing as the foundation for future success in the field. Importantly, a collaborative and well-trained community, including a new generation of young scholars, will be able to communicate the societal impacts of engineering synthetic cells and organelles responsibly and effectively to the public. Particularly with respect to questions of how to share intellectual property to benefit humanity while continuing to reward innovation, biosafety, biosecurity and other unique ethical and philosophical considerations, including the most fundamental question of all: 'what is life?'.

### Acknowledgements
The authors thank all the speakers and participants of the SynCell2020/21 and associated activities for their critical input, inspiring discussions and engaged participation. The National Science Foundation (CBET-1841170), the Max Planck Society, the New Mexico Consortium and the University of New Mexico are acknowledged for their financial support.

**Oskar Staufer** is in the Max Planck Institute for Medical Research and the Max Planck School Matter to Life, both in Heidelberg, Germany, and the Max Planck Bristol Center for Minimal Biology, University of Bristol, Bristol, United Kingdom
oskar.staufer@mr.mpg.de
http://orcid.org/0000-0002-8015-3132
**Jacqueline A De Lora** is in the Max Planck Institute for Medical Research, Heidelberg, Germany
http://orcid.org/0000-0001-5599-7838
**Eleonora Bailoni** is at the University of Groningen, Groningen, Netherlands

**Alisina Bazrafshan** is at Emory University, Atlanta, United States
http://orcid.org/0000-0002-3259-8196
**Amelie S Benk** is in the Max Planck Institute for Medical Research, Heidelberg, Germany
**Kevin Jahnke** is in the Max Planck Institute for Medical Research, Heidelberg, Germany
http://orcid.org/0000-0001-7311-6993
**Zachary A Manzer** is at Cornell University, Ithaca, United States
http://orcid.org/0000-0002-8225-8990
**Lado Otrin** is in the Max Planck Institute for Dynamics of Complex Technical Systems, Magdeburg, Germany
http://orcid.org/0000-0001-5862-456X
**Telmo Díez Pérez** is at the University of New Mexico, Albuquerque, United States
**Judee Sharon** is at the University of Minnesota, Minneapolis, United States
http://orcid.org/0000-0001-5691-0407
**Jan Steinkühler** is at Northwestern University, Evanston, United States
http://orcid.org/0000-0003-4226-7945
**Katarzyna P Adamala** is at the University of Minnesota, Minneapolis, United States, and is also a member of the Build-a-Cell Research coordination network
http://orcid.org/0000-0003-1066-7207
**Bruna Jacobson** is at the University of New Mexico, Albuquerque, United States
**Marileen Dogterom** is at the Delft University of Technology, and the BaSyC Consortium, both in Delft, Netherlands
http://orcid.org/0000-0002-8803-5261
**Kerstin Göpfrich** in the Max Planck Institute for Medical Research and the Max Planck School Matter to Life, both in Heidelberg, Germany
http://orcid.org/0000-0003-2115-3551
**Darko Stefanovic** is at the University of New Mexico, Albuquerque, United States
**Susan R Atlas** is at the University of New Mexico, Albuquerque, United States
http://orcid.org/0000-0003-1542-2700
**Michael Grunze** in the Max Planck Institute for Medical Research and the Max Planck School Matter to Life, both in Heidelberg, Germany
http://orcid.org/0000-0002-2335-9513
**Matthew R Lakin** is at the University of New Mexico, Albuquerque, United States
**Andrew P Shreve** is at the University of New Mexico, Albuquerque, United States
http://orcid.org/0000-0002-9567-3181
**Joachim P Spatz** is in the Max Planck Institute for Medical Research, the Max Planck School Matter to Life, and Heidelberg University, all in Heidelberg, Germany, and the Max Planck Bristol Center for Minimal Biology, University of Bristol, Bristol, United Kingdom
Joachim.Spatz@mpimf-heidelberg.mpg.de
http://orcid.org/0000-0003-3419-9807

Gabriel P López is at the University of New Mexico, Albuquerque, United States, and is also a member of the Build-a-Cell Research coordination network gplopez@unm.edu

http://orcid.org/0000-0002-5383-0708

*Author contributions:* Oskar Staufer, Conceptualization, Project administration, Visualization, Writing – original draft, Writing – review and editing; Jacqueline A De Lora, Conceptualization, Writing – review and editing; Eleonora Bailoni, Conceptualization, Writing – review and editing; Alisina Bazrafshan, Conceptualization, Writing – review and editing; Amelie S Benk, Conceptualization, Writing – review and editing; Kevin Jahnke, Conceptualization, Writing – review and editing; Zachary A Manzer, Conceptualization, Writing – review and editing; Lado Otrin, Conceptualization, Writing – review and editing; Telmo Díez Pérez, Conceptualization, Project administration; Judee Sharon, Conceptualization, Writing – review and editing; Jan Steinkühler, Conceptualization, Writing – review and editing; Katarzyna P Adamala, Conceptualization, Writing – review and editing; Bruna Jacobson, Conceptualization, Writing – review and editing; Marileen Dogterom, Conceptualization, Writing – review and editing; Kerstin Göpfrich, Conceptualization, Writing – review and editing; Darko Stefanovic, Conceptualization; Susan R Atlas, Conceptualization, Funding acquisition, Project administration, Visualization, Writing – original draft, Writing – review and editing; Michael Grunze, Conceptualization, Funding acquisition, Writing – review and editing; Matthew R Lakin, Conceptualization, Project administration, Visualization, Writing – original draft, Writing – review and editing; Andrew P Shreve, Conceptualization, Funding acquisition, Project administration, Writing – original draft, Writing – review and editing; Joachim P Spatz, Conceptualization, Funding acquisition, Project administration, Supervision, Writing – review and editing; Gabriel P López, Conceptualization, Funding acquisition, Project administration, Visualization, Writing – original draft, Writing – review and editing

*Competing interests:* The authors declare that no competing interests exist.

## Funding

| Funder | Grant reference number | Author |
|---|---|---|
| National Science Foundation | CBET-1841170 | Gabriel P López |
| Max Planck Society | Max Planck School Matter to Life | Joachim P Spatz |
| New Mexico Consortium | | Gabriel P López |

The funders had no role in study design, data collection and interpretation, or the decision to submit the work for publication.

**Decision letter and Author response**
Decision letter https://doi.org/10.7554/eLife.73556.sa1
Author response https://doi.org/10.7554/eLife.73556.sa2

# Additional files

## Supplementary files

• Supplementary file 1. Supplementary file providing a summary of the SynCell 2021 lecture series, oral and poster contributions as well as quotations from early-career panelists. Table S1: Presentations at SynCell2020. Table S2: Presentations at the SynCell2021 Spring Lecture Series with Build-A-Cell. Table S3: Featured and Contributed Oral Presentations at SynCell2021. Table S4: Poster Presentations at SynCell2021. Table S5: Lightning Talks at SynCell2021 (Presenters chosen by jury from poster presenters). Box S1: Quotations from the early-career panelists of the SynCell20/21 workshop on the future challenges in the field.

• Transparent reporting form

## Data availability

This Feature Article does not contain any primary data.

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
