## [Decision Letter]

**Decision letter after peer review:**

Thank you for submitting your article "Building a community to engineer synthetic cells and organelles from the bottom-up" to *eLife* for consideration as a Feature Article. Your article has been reviewed by 3 peer reviewers, and the evaluation has been overseen by 2 members of the *eLife* Features Team (Helga Groll and Peter Rodgers). The following individuals involved in review of your submission have agreed to reveal their identity: Atul Parikh, Erdinc Sezgin

The reviewers and editors have discussed the reviews and we have drafted this decision letter to help you prepare a revised submission.

Summary:

This is a nicely written article that summarizes the proceedings of the SynCell 2020/2021 conference, which brought together a diverse group of early-stage and senior researchers focused on designing, manipulating and creating cell-like synthetic activities.

The report highlights the importance of building an international and multidisciplinary community dedicated to developing the fundamental understanding and technical know-how to create synthetic cells from the bottom up. It discusses achievements within the emerging field, projects broader impacts, including socioeconomic and technological ones, and identifies challenges and opportunities. It concludes with a call for building an integrated research community through resource-sharing, enhanced communication, and cross-institutional and transnational educational initiatives.

The fascinating convergence of scientific and technological knowledge with philosophical and ethical considerations will inspire early-career scholars and bring transformative innovations to society.

Essential revisions:

1. To an engaged and informed readership, the article, would serve as a valuable resource and a roadmap if the conference proceedings (a) identified specific milestones, (b) highlighted major accomplishments, and (c) tabulated specific challenges and opportunities that must be addressed to meet the aspirations of the field.

A graphic or a table highlighting the timeline/milestones/opportunities and projecting the future would prove useful.

2. The description of the perceived limitations of the approach needs elaborating. Many features of complex systems are often lost when complexities are reduced and systems dissected to their presumed essential ingredients. But which emergent aspects are important? What does minimal complexity entail? How does minimal complexity change with the increased functionality, adaptability and survivability?

3. The discussion in the role of function in guiding the cellular design appears to be conspicuously missing. What ingredients must be incorporated to recapitulate some of the most basic abilities of the living cell, such as surviving in harsh environments, processing physical and chemical signals from their local environment, making decisions, i.e., selecting a specific phenotype (e.g., gene expression, morphology, organization) from a range of possibilities in response to an extracellular change, or identifying signals from noisy backgrounds?

4. The discussion on hierarchical organization is particularly interesting. I recommend considering insights from the growing body of literature on eukaryogenesis, including possibly a description of the initiative from the Simons-Moore Foundation.

5. It would be important to contemplate on the inclusivity aspect of the field while providing a perspective on the future of this discipline. Almost all research areas are dominated by a few established people or family-like consortiums – mostly from established countries, which makes it difficult for the young researchers to be part of the community. I cannot say this is the case for this effort authors are presenting here, but it needs to be kept in mind. Clearly, a few programs (such as Max Planck programs) have taken the initiative for this endeavour, but future steps would be stronger if more parties are involved regardless of their geographical whereabouts, career stage, or previous involvement in the established network.

---

## [Author Response]

Essential revisions:1. To an engaged and informed readership, the article, would serve as a valuable resource and a roadmap if the conference proceedings (a) identified specific milestones, (b) highlighted major accomplishments, and (c) tabulated specific challenges and opportunities that must be addressed to meet the aspirations of the field.A graphic or a table highlighting the timeline/milestones/opportunities and projecting the future would prove useful.

We thank the reviewers for bringing this point to our attention and we agree that these are important aspects to be considered in the community. Several very complete reviews have been published on these matters recently. For instance, Gallup *et al.,* tabulate in an up-to-date review the 10 future challenges in the field and Meng *et al.,* have extensively reviewed and illustrated the achievements of the field in the last decade. However, the major focus of our Feature Article lies inidentifying the infrastructural, educational and collaborative bottlenecks and opportunities in the field. In the view that the aspects raised by the reviewers are of importance for an informed readership, we now direct the reader to the reviews mentioned above in the revised version of our manuscript, and write (page 8, line 6):

“Many of these were highlighted at the recent SynCell2020/21 conference (see Supplementary Box 2 and Supplementary Information for listings of the conference program and links to recordings) and have been extensively reviewed recently by Gallup et al., and Meng et al.,^7,32^.”

Moreover, we now provide an additional supplementary figure in the revised version of our manuscript that summarizes quotes collected from early career panelists of the SynCell2021 workshop on specific challenges in the field.

2. The description of the perceived limitations of the approach needs elaborating. Many features of complex systems are often lost when complexities are reduced and systems dissected to their presumed essential ingredients. But which emergent aspects are important? What does minimal complexity entail? How does minimal complexity change with the increased functionality, adaptability and survivability?

We agree with the reviewers that our article would benefit from a more balanced description of the systematic limitations of the reductionist approach. In order to draw the reader’s attention to such issues, we have now added one paragraph to our revised manuscript that details two systematic limitations underlying the field and the constraints they entail (page 11, line 4):

“Although the reductionist approach pursued in the field of engineering synthetic cells and organelles has proven to be powerful for several lines of research, it is also confronted with systematic limitations. Foremost, as an engineering approach that iteratively reduces the complexity of living systems, the construction of synthetic cells will always be subjected to the problem of “hidden variables''^55^. Unobservable or unidentified molecular components of importance to natural systems might be underrepresented or absent in the recreated in vitro system. As such variables are by definition unknown and can contribute to the stochastic behavior of a system, biological phenomena could potentially be masked behind noise effects in synthetic cells and organelles. Moreover, living cells can not only be considered as the simple sum of their parts. Emergence in living systems is poorly defined and understood, and the possibility to capture this aspect by a modular approach remains to be evaluated. A second systematic limitation of the reductionist approach lies in the fact that reduced complexity does not directly imply increased understanding of a biological phenomenon. This is well illustrated by the creation of minimal bacterial genomes with only 473 genes^23^. Despite the remarkably reduced complexity and size of this genome, 149 of the genes are still of unknown function. Addressing these limitations provides opportunities for ongoing research and collaborative efforts, even as progress continues in advancing understanding and functional applications of synthetic cells and organelles.”

3. The discussion in the role of function in guiding the cellular design appears to be conspicuously missing. What ingredients must be incorporated to recapitulate some of the most basic abilities of the living cell, such as surviving in harsh environments, processing physical and chemical signals from their local environment, making decisions, i.e., selecting a specific phenotype (e.g., gene expression, morphology, organization) from a range of possibilities in response to an extracellular change, or identifying signals from noisy backgrounds?

We agree with the reviewers that providing a description of basic cellular functions, especially those associated with signal integration and sensing, would further highlight the challenges of synthetic cell engineering and modular aspects of the approach. We therefore now write on page 3, line 25 of our revised manuscript:

“Cellular functions entail a diverse spectrum of modular components that regulate cell behavior. Such cellular functions include but are not limited to the ability to process chemical and physical signals from extracellular stimuli (information processing and decision making), cellular motions and adaptation to the environment (molecular adaptivity), cellular replication through division (proliferation) and nutrient uptake and garbage disposal (energy homeostasis). These abilities have helped to guide the functional design of synthetic cells and their equipment with modules that allow for their interaction with the extracellular environment and discrimination of extracellular signals. For instance, extracellular morphogen sensing systems have been implemented in protocells that allow for structural adaptation in a reaction-diffusion model and discrimination between uni-directional and counter-directional morphogen gradients11. Moreover, tactile behaviour in the form of chemo- and phototaxis has been engineered into synthetic cell surrogates^12,13^.”

4. The discussion on hierarchical organization is particularly interesting. I recommend considering insights from the growing body of literature on eukaryogenesis, including possibly a description of the initiative from the Simons-Moore Foundation.

We appreciate the suggestion of the reviewer and agree that eukaryogenesis, as well as other related topics such as origin-of-life research, are closely related to questions addressed in the current submission. Indeed, many of the participants in SynCell2020/21 are active researchers in these and other related areas. Although an extensive discussion of recent advances in these areas extends beyond the scope or our submission, we have made revisions to highlight the existence of these connections and also the importance of foundation support. To that end, the third-to-last paragraph now reads:

“For all the proposed measures, commitment and support from funding bodies, political and regulatory authorities as well as universities with established teaching infrastructure will be essential. This is most important for the successful installation of a strategic open-source platform for synthetic biology and student exchange programs, such as now established between the University of New Mexico and the Max Planck Society. We also observe that connections to and inspiration drawn from other research communities will be important. For example, research advances addressing origin-of-life questions, investigation of basic principles of life, and exploration of eukaryogenesis connect to many scientific themes that arise in the study of synthetic cells and organelles. This connection is highlighted by the observation that many SynCell2020/21 participants are also active in these related communities. Furthermore, expansion of the research community in synthetic cells to connect to these and other related scientific communities opens additional opportunities for research support, including that available from private and philanthropic foundations. Recent examples of such initiatives include the "Life?" program championed by the Volkswagen Foundation^57^ and the "Project on the Origin of the Eukaryotic Cell," sponsored jointly by the Gordon and Betty Moore Foundation and the Simons Foundation^58^.”

5. It would be important to contemplate on the inclusivity aspect of the field while providing a perspective on the future of this discipline. Almost all research areas are dominated by a few established people or family-like consortiums – mostly from established countries, which makes it difficult for the young researchers to be part of the community. I cannot say this is the case for this effort authors are presenting here, but it needs to be kept in mind. Clearly, a few programs (such as Max Planck programs) have taken the initiative for this endeavour, but future steps would be stronger if more parties are involved regardless of their geographical whereabouts, career stage, or previous involvement in the established network.

While we agree that there are significant inequities in accessibility to the field of synthetic cells and organelles faced by aspiring researchers across the world (as there are in other emerging fields), we believe that the SynCell2020/21 symposium activities and this manuscript are precisely a significant effort to address these inequities. What initially was envisioned as a somewhat exclusive in-person conference (with 150 attendees) in a somewhat remote location (Santa Fe, NM), was, by necessity engendered by the pandemic, transformed into a widely-advertised, free, virtual conference with a huge number of participants (~750) from across 6 continents. As indicated in the manuscript, the symposium provided several activities specifically geared toward young participants including tutorials, poster sessions and, importantly, a post-conference workshop that formed the basis for the manuscript itself, for which one half of the 22 coauthors are young researchers (i.e., students or postdocs). A major component of the workshop was a session specifically designed to elicit creative new ideas for expanding the international reach of educational and research opportunities, while advancing the field and its technological capabilities in ways that resonate with a new generation. The symposium activities and this manuscript (including the index of available online recordings) raise awareness of global efforts and networks dedicated to synthetic cell research; these resources further highlight ongoing activities in which researchers outside these networks can participate. Notably, as a direct result of the pandemic, there are now numerous online synthetic cell colloquia and seminar series, highlighted by SynCell2020/21 speakers, that are readily accessed from anywhere in the world live or as YouTube recordings. We therefore feel strongly that our manuscript does indeed contemplate inclusivity as well as a vision of the future of the discipline by describing activities geared toward engaging young, ethnically diverse, and geographically diverse researchers. Indeed, this was a major motivating factor for our submission of this manuscript to *eLife*, an open access journal. We have made several additions to the manuscript text to emphasize our vision to enhance diversity, equity and inclusion in the emerging field of synthetic cells and organelles.

We thank the reviewers for sharing our mutual concern for inclusion as the field of synthetic cells and organelles matures; it is incumbent upon us (as the reviewers suggest) to continue to strive for accessibility of this field to researchers regardless of their institutional locale, professional stage or previous connections. We are dedicated to this task and several of us are already engaging in the organization of SynCell2022, for which we are advocating for the retention of the virtual component, as well as planning fundraising activities to enable widespread virtual attendance and in-person attendance of diverse, young researchers. In this manner, we hope to instill a culture of accessibility and inclusion in this important emerging field.